# The 26S Proteasome Switches between ATP-Dependent and -Independent Mechanisms in Response to Substrate Ubiquitination

**DOI:** 10.3390/biom12060750

**Published:** 2022-05-26

**Authors:** Abramo J. Manfredonia, Daniel A. Kraut

**Affiliations:** Department of Chemistry, Villanova University, Villanova, PA 19085, USA; amanfre6@villanova.edu

**Keywords:** ubiquitin, ubiquitin–proteasome system, proteasome, ATP-dependent protease, protein degradation

## Abstract

The ubiquitin–proteasome system is responsible for the bulk of protein degradation in eukaryotic cells. Proteins are generally targeted to the 26S proteasome through the attachment of polyubiquitin chains. Several proteins also contain ubiquitin-independent degrons (UbIDs) that allow for proteasomal targeting without the need for ubiquitination. Our laboratory previously showed that UbID substrates are less processively degraded than ubiquitinated substrates, but the mechanism underlying this difference remains unclear. We therefore designed two model substrates containing both a ubiquitination site and a UbID for a more direct comparison. We found UbID degradation to be overall less robust, with complete degradation only occurring with loosely folded substrates. UbID degradation was unaffected by the nonhydrolyzable ATP analog ATPγS, indicating that UbID degradation proceeds in an ATP-independent manner. Stabilizing substrates halted UbID degradation, indicating that the proteasome can only capture UbID substrates if they are already at least transiently unfolded, as confirmed using native-state proteolysis. The 26S proteasome therefore switches between ATP-independent weak degradation and ATP-dependent robust unfolding and degradation depending on whether or not the substrate is ubiquitinated.

## 1. Introduction

The ubiquitin–proteasome system (UPS) is responsible for the bulk of protein degradation in eukaryotic cells [1]. The 26S proteasome is composed of two subunits: the 19S regulatory particle (RP) and the 20S core particle (CP) [2,3]. The 19S RP can be further divided into two subcomplexes: the base, which is responsible for translocation and unfolding of proteins; and the lid, which acts as a scaffold for targeted protein engagement and deubiquitination. The base subunit is primarily composed of six distinct ATPase subunits (Rpt1–6) that work to mechanically unfold and translocate proteins using the energy from ATP hydrolysis [4]. The 20S CP has a barrel-like structure composed of two sets of α and β rings with distinct peptidase sites, forming a proteolytic core closed by a “gate” formed from the N termini of the α subunits. Docking of the 19S ATPases to the 20S CP opens this gate, activating the CP’s peptidase activity [2,3].

Proteins are typically targeted to the proteasome via polyubiquitination [1,5]. Ubiquitin is activated by an E1 enzyme and transferred to an E2 enzyme. An E3 enzyme binds to both the E2 and a degron region in the target protein, and ubiquitin is transferred to a lysine residue in either the target or in an already attached ubiquitin. The latter yields the formation of polyubiquitin chains, whose varying and complex architecture determines the fate of the ubiquitinated protein [6]. In addition, a small number of known proteins are targeted to the 26S proteasome independent of ubiquitin [7]. These proteins can self-target to the proteasome via a ubiquitin-independent degron (UbID), typically an unstructured domain or a domain containing unstructured regions. The mechanism by which UbIDs are recruited to the proteasome is unclear, in part due to a lack of sequence conservation between UbIDs. In yeast, there are two well-characterized UbID-containing proteins; Rpn4 and ornithine decarboxylase (ODC) [8,9].

Rpn4 is a transcription factor responsible for regulating proteasome subunit expression levels. The native protein contains both ubiquitin-dependent and -independent degrons for proteasomal targeting, allowing Rpn4 to participate in a negative feedback loop to maintain proteasome homeostasis [8]. The N-terminal 80 residues of Rpn4 serve as a portable UbID degron that, when fused to other proteins, converts them into proteasome substrates. The first ~20 residues are intrinsically disordered, providing a potential basis for proteasome engagement. This degron interacts with 19S lid subunits, though the mechanism of recruitment and downstream effect on degradation is still unclear. Rpn4 can also be ubiquitinated, resulting in either ubiquitin-independent or -dependent degradation, depending on the cellular context [10].

ODC is an enzyme that catalyzes the committed step in the polyamine synthesis pathway. ODC degradation is regulated by UbIDs in both yeast and mammals, although the degron sequence and positioning differ between species [9,11]. The N-terminal 44 residues of yeast ODC (yODC) are portable and sufficient to induce ubiquitin-independent degradation in target substrates [9]. The yODC UbID is an unstructured domain, again providing a potential proteasomal initiation site. Replacement of the UbID with an alternative unstructured sequence is sufficient to maintain ubiquitin-independent degradation by the 26S proteasome, highlighting the lack of sequence conservation amongst UbIDs [9]. 

The proteasome processively unfolds and degrades substrates, but each time a folded domain is encountered, there is an opportunity for the proteasome to stall and release the substrate. We have previously shown that UbID substrates are degraded with lower processivity than ubiquitinated substrates, but the mechanisms underlying these differences remain unclear [12,13]. Herein we show that the proteasome degrades UbID substrates using an ATP-independent mechanism, leading to a lower ability to unfold substrates.

## 2. Materials and Methods

### 2.1. Constructs

The N-terminal 44 residues of yODC and the N-terminal 80 residues of Rpn4 were each amplified from yeast genomic DNA and individually cloned into pCMH39 (His-SUMO-R-Neh2Dual-Barnase∆K-C-DHFRkΔC) using Gibson Assembly to replace the R-Neh2Dual domain [14]. Barnase was destabilized (L89G [13]) and a PPPY Rsp5 binding site was added after the degron by oligo-directed mutagenesis to create plasmids pAJM1 (His-SUMO-yODC^1–44^-PPXY-BarnaseL89G-C-DHFRkΔC) and pAJM2 (His-SUMO-Rpn4^1–80^-PPXY-BarnaseL89G-C-DHFRkΔC). 

### 2.2. Substrate Purification

Rpn4- and yODC-containing substrates were overexpressed, purified (including removal of the His-SUMO tag), labeled with sulfo-cyanine5 maleimide (Lumiprobe, Hunt Valley, MD, USA), and repurified by gel filtration as described previously [14]. UBL-sGFP-102-His_6_ was purified as described previously [15]. Sequences and molecular weights are given in Appendix A.

### 2.3. Substrate Ubiquitination 

Cy5-labeled substrates were ubiquitinated using the Rsp5 ubiquitination system. Reactions contained 166 nM E1, 2.9 μM UbcH7 (E2), 2.9 μM Rsp5 (E3), 1.33 mg/mL ubiquitin, 4 mM ATP, 5 μM substrate, 500 μM NADPH, and 1 μM DTT in Rsp5 ubiquitination buffer (25 mM Tris, 50 mM NaCl, 4 mM MgCl_2_ at pH 7.5). Reactions were incubated for 2 h at 30 °C and purified by spin-size exclusion as described previously [14].

### 2.4. Proteasome Purification 

Proteasome was purified from strain YYS40 [16] using a 3×-FLAG-tagged copy of the Rpn11 subunit of the 19S regulatory particle as described previously [14].

### 2.5. Degradation Assays

Degradation assays were conducted with 20 nM Cy5-labeled substrate (or UBL-sGFP-102-His_6_) and 100 nM WT proteasome over a 0.5–4 h time course as described previously [14]. Assays were carried out in degradation buffer (50 mM TrisCl, 5 mM MgCl_2_, 5% (*v*/*v*) glycerol, and 0.1% (*v*/*v*) Tween-20 at pH 7.5) with 1 mM ATP, 10 mM creatine phosphate, 0.1 mg/mL creatine kinase, 1 mg/mL BSA, 2 mM DTT, and 1% (*v*/*v*) DMSO at 30 or 10 °C. Reactions also contained 500 µM NADPH to stabilize DHFR as indicated. Assays in the presence of ATPγS (1 mM) did not contain the ATP regeneration system (ATP, creatine phosphate, and creatine kinase). Reactions were analyzed via SDS-PAGE and fluorescence detection using a Typhoon FLA 9500 (Cytiva, Marlborough, MA, USA). For reconstitution assays, 19S and 20S proteasomes, purified as described previously, were reconstituted in a 3:1 ratio (75 nM:25 nM) for 5 min at 30 °C before beginning the assay [15]. Samples from reconstitution assays were also removed to ice at the indicated times, run on a 3.5% native gel, and visualized using Suc-LLVY-AMC in the presence of 0.02% SDS [17].

### 2.6. Native State Proteolysis

Proteolysis assays were carried out over 16 min time courses at 30 or 10 °C. Reactions contained 80 nM substrate, 0.0125 mg/mL thermolysin, 1 mg/mL BSA, 2 mM DTT, and 1% (*v*/*v*) DMSO in degradation buffer (50 mM TrisCl, 5 mM MgCl_2_, 5% (*v*/*v*) glycerol, and 0.1% (*v*/*v*) Tween-20 at pH 7.5). Reactions also contained 500 µM NADPH to stabilize DHFR as indicated. Reactions were quenched in 120 mM EDTA and analyzed via SDS-PAGE and fluorescence detection using a Typhoon FLA 9500. 

## 3. Results

### 3.1. Ubiquitin-Mediated Degradation Is More Robust Than Ubiquitin-Independent Degradation

To investigate differences between ubiquitin-independent and -dependent degradation, we designed substrates that could be degraded by either ubiquitin-dependent or -independent pathways in our previously described unfolding ability assay (Figure 1A) [12,13,14,18]. The substrates contained an N-terminal ubiquitin-independent degron (UbID) derived from the N-terminus of either yeast Rpn4 (containing four lysines to serve as potential ubiquitination sites) or yeast ornithine decarboxylase (yODC containing three lysines), an Rsp5 binding site (PPPY) to allow ubiquitination [16], an easy-to-unfold lysine-free barnase domain (destabilized via mutation), and finally, a difficult-to-unfold C-terminal DHFR domain (with all but one lysine removed to prevent internal ubiquitination). Substrates also contained a single cysteine between the barnase and DHFR domains for labeling with Cy5 and fluorescence detection. The proteasome unfolds and degrades the degron and barnase domain of the substrate, eventually encountering the difficult-to-unfold DHFR region. The proteasome may then either stall and irreversibly release the DHFR fragment, or successfully unfold and degrade it. The unfolding ability (U) is described by the ratio of the rate of unfolding and degradation of the substrate to the rate of release of fragment (k_deg_^frag^/k_rel_^frag^ in Figure 1A), and can be determined from the extent of fragment formation versus complete degradation [14]. Assays were conducted either in the absence or presence of NADPH, which stabilizes DHFR [19]. The proteasome is not directly affected by NADPH, as degradation of a GFP-containing substrate lacking DHFR was unaffected by NADPH (Appendix A).

Ubiquitinated-Rpn4-containing substrate had an unfolding ability of 2.1 ± 0.3 in the presence of NADPH, meaning it was completely degraded in 68% ± 3% of engagements with the proteasome, with stalling and release of a DHFR-containing fragment the other ~32% of the time. The ubiquitinated substrate was completely degraded in the absence of NADPH (1% ± 1% of engagements led to fragment formation), which stabilized the DHFR domain (Figure 1B,C, Appendix A). The nonubiquitinated-Rpn4-containing substrate was unable to be completely degraded in the presence of NADPH, with essentially all engagements leading to stalling and release of the DHFR domain. Nonetheless, in the absence of NADPH, there was substantial complete degradation (U = 2.1 ± 0.3), indicating that a folded domain (DHFR) can still be unfolded and degraded 67% ± 3% of the time by the proteasome via UbID targeting (Figure 1D,E, Appendix A). Degradation of both ubiquitinated and nonubiquitinated substrates was due to the proteasome, as the reactions were proteasome-dependent and were substantially slowed by proteasome inhibitors (Appendix A). The proteasome degraded the ubiquitinated substrate about three times faster than the nonubiquitinated substrate, indicating that binding, engagement, or unfolding and translocation of UbID substrates was less efficient. Replacing the Rpn4 UbID with that from yODC gave similar results, suggesting that ubiquitin-independent degradation is universally less robust than ubiquitin-dependent degradation (Appendix A), and is able to degrade loosely folded but not tightly folded proteins. Some loosely folded proteins can be degraded without the assistance of proteasomal motor proteins [20]. To determine whether degradation of UbID substrates was ATP-dependent, we attempted to stall degradation using the nonhydrolyzable ATP analog ATPγS.

### 3.2. Ubiquitin-Independent Degradation Is ATP-Independent 

ATPγS causes the proteasome to stall in s3-like substrate processing states [21], preventing switching into the s1 substrate-accepting state and thus halting proteasomal engagement and degradation. For some substrates, stalling allows an ATP-independent clipping mechanism to predominate, and the ATPγS-stalled proteasome can also degrade some intrinsically disordered proteins [15,22]. ATPγS considerably slowed degradation of the ubiquitinated-Rpn4-containing substrate and reduced the unfolding ability from 2.1 ± 0.3 to 0.6 ± 0.2 (64% ± 6% of engagements stalled and led to fragment formation) in the presence of NADPH. In the absence of NADPH, degradation remained slow, but the DHFR domain was largely or completely degraded (U = 6 ± 2; 14 ± 4% of engagements stalled and led to fragment formation), suggesting that DHFR can be degraded in an ATP-independent manner as long as it is not stabilized by NADPH (Figure 2A,B). We also saw that a small amount (~10%) of the ubiquitinated substrate was deubiquitinated instead of being degraded (Figure 2A,B, Appendix A). The non-ubiquitinated Rpn4 substrate was almost unaffected by ATPγS (Figure 2C,D), indicating that UbID substrates were degraded primarily or exclusively in an ATP-independent manner. Without the ability to hydrolyze ATP, the ubiquitin-dependent and -independent substrates degraded in a similar fashion, suggesting a “default” ATP- and ubiquitin-independent degradation mode that is still capable of degrading folded proteins such as DHFR as long as they are not excessively stable. We obtained similar results when using the yODC-containing substrate (Appendix A). To ensure that degradation was due to the 26S proteasome, and not to trace amounts of free 20S in our assay, we reconstituted the 26S proteasome using 20S and excess 19S proteasome and obtained similar results (Appendix A). Therefore, the 26S proteasome is able to degrade loosely folded proteins in an ATP-independent fashion.

### 3.3. Transient Unfolding of Substrates Is Required for Complete Degradation

Typically, substrate engagement and translocation by the proteasome are driven by the hexameric ATPase ring. The ATPases actively unfold engaged proteins for subsequent cleavage. It is possible the proteasome is capturing substrate in a transiently unfolded state or that the substrate is natively unfolded, allowing for this “default” ATP- and ubiquitin-independent mechanism to function. To investigate these models, we performed proteolysis assays using thermolysin, a nonspecific protease only capable of cleaving unfolded regions of proteins. Assays were carried out using nonubiquitinated-Rpn4-containing substrate in the presence and absence of NADPH and with FITC-casein, a natively unfolded protein. Reactions were initiated with the addition of thermolysin, and results were visualized via fluorescence detection after SDS-PAGE analysis. 

Thermolysin was able to degrade FITC-Casein around five times faster than nonubiquitinated-Rpn4-containing substrate without NADPH present, indicating the substrate was likely loosely folded, not natively unfolded. The addition of NADPH stabilized the substrate sufficiently so that there was much less degradation occurring, with stable DHFR-containing fragment persisting throughout the time course (Figure 3A). Similar results were obtained for yODC-containing substrates (Appendix A). When native state proteolysis was performed at 10 °C, FITC-casein was still completely degraded, while nonubiquitinated-Rpn4-containing substrate was significantly less degraded, with stable barnase-DHFR-containing fragment persisting (Figure 3B). Thermolysin is likely unable to degrade the substrate because there is little transient unfolding occurring at lower temperatures. Unfolding ability assays performed in the absence of NADPH at 10 °C showed little degradation of nonubiquitinated-Rpn4-containing substrate, while ubiquitinated-Rpn4-containing substrate was still processively degraded (Figure 3C,D). This result shows that the proteasome likely engages UbID substrates by exploiting these transiently unfolded states rather than by actively unfolding the protein.

## 4. Discussion

Our results indicate that proteasomal degradation is less robust and processive when substrates are targeted via a UbID compared with polyubiquitination. The proteasome was unable to degrade tightly folded UbID-containing substrates, and could only degrade loosely folded domains. Furthermore, ubiquitin-independent degradation was unaffected by the presence of ATPγS, indicating that UbID processing is ATP-independent (Figure 4).

Whereas the 26S proteasome is typically responsible for degrading ubiquitinated proteins, the 20S CP on its own has been associated with ATP-independent degradation of smaller and unfolded proteins [23,24]. The 20S CP has a narrow pore with a gate that is typically closed, preventing access by proteins, but unfolded proteins may be able to open this gate [25]. The 19S RP binds to the 20S and opens the gate in addition to its other roles in ubiquitin recognition, unfolding, and translocation [2]. Previous studies showed the 19S is required for degradation of Rpn4 via its UbID, and at least the base subcomplex of the 19S is required for degradation of yODC [8,9]. Therefore, the 19S might be playing a role in UbID degradation simply by opening the gate and thereby allowing unfolded portions of the substrate into the central channel. Preventing transient unfolding, either by addition of a stabilizing ligand or lowered temperature, prevented ubiquitin-independent proteasomal degradation of our substrates, further supporting the model that proteins must unfold on their own before UbID degradation is possible (Figure 4B). 

Proteasomes can exist in either a substrate accepting (s1) or substrate translocating (s3-like) conformation, and ATPγS, which supported UbID degradation, promotes s3-like conformations [3,21]. In these substrate-translocating conformations, the 19S pore is fully aligned with the 20S channel, providing a potential pathway for substrates to traverse into the 20S. Alternatively, it is possible that the substrate bypasses the 19S pore and enters via transient openings or fluctuations at the 19S–20S interface (Figure 4B). Once substrates are engaged, translocation is typically driven by ATP hydrolysis, so the mechanism by which UbID-containing substrates are pulled into the 20S core remains unexplained. Recent studies suggest the ß proteases of the 20S CP are able to exert a small “tugging” force upon peptide bond cleavage, providing a potential mechanism for ATP-independent translocation of UbID-containing substrates [25]. 

The two UbID’s examined herein, from Rpn4 and yODC, have no obvious sequence conservation with one another. Furthermore, there is no apparent evolutionary conservation of these sequences, as demonstrated by the lack of any homologs beyond budding yeast when searching using just the UbID in BLAST. Given that the yODC sequence can also be replaced with an unrelated unstructured sequence [9], it seems likely that these UbID domains are simply unstructured regions that allow for proteasomal capture of a poorly folded or transiently unfolded substrate. The presence of terminal unstructured regions correlates with a short protein half-life in both yeast and mammals [26]. One possibility is that UbID degradation is a vestige of a preubiquitin world, when, in the absence of the extensive protein quality control system that has evolved in eukaryotes, poorly folded proteins might have been targeted to the proteasome simply based on their folding properties. In the case of Rpn4, it has been suggested that UbID degradation is a failsafe allowing downregulation of the proteasome even under circumstances where general UPS activity is compromised [10]. UbID degradation may then more generally serve as a backup mechanism that allows degradation of the proteins most likely to aggregate under high stress.

## Figures and Tables

**Figure 1 biomolecules-12-00750-f001:**
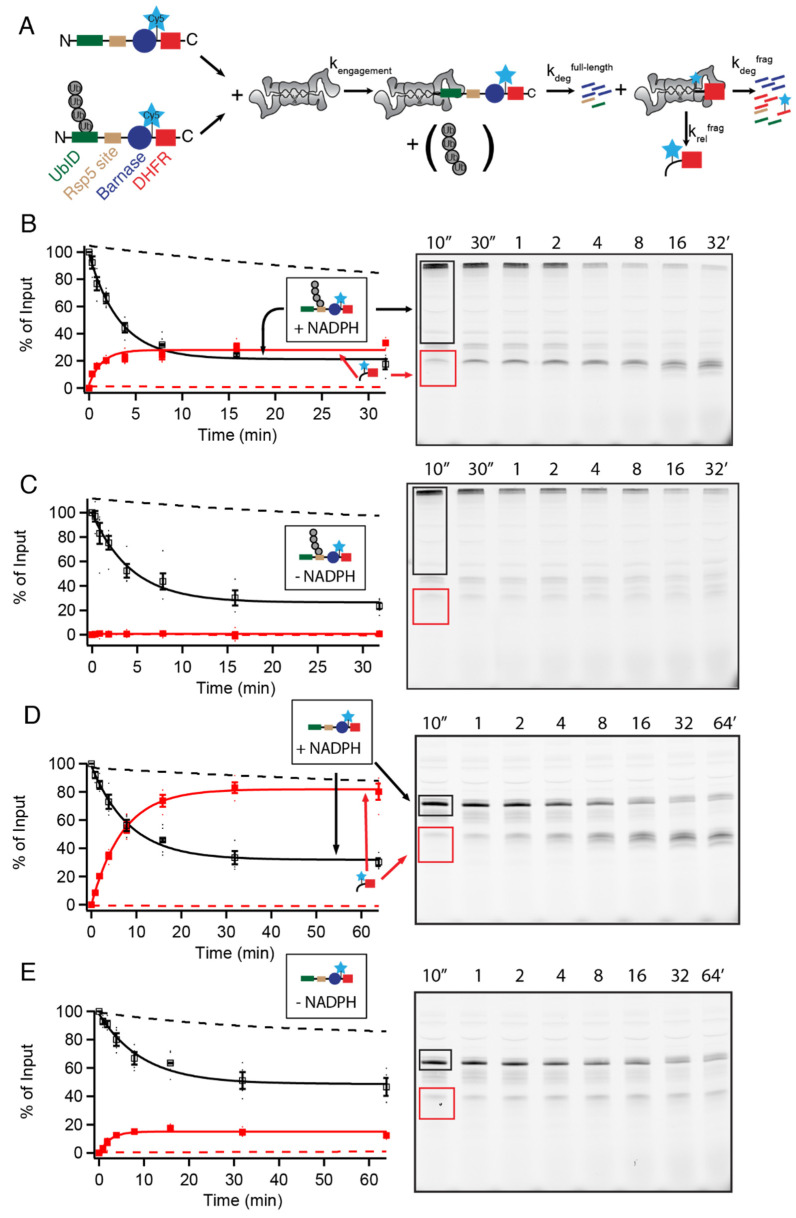
Proteasomal unfolding ability depends on targeting mechanism and substrate stability. (**A**) Unfolding ability assay where proteasome degradation proceeds with the engagement of ubiquitinated or nonubiquitinated substrate (k_engagement_), unfolding and degrading of barnase (k_deg_^full-length^), and then either irreversible release (k_rel_^frag^) or degradation (k_deg_^frag^) of DHFR. The ratio of k_deg_ to k_rel_ describes the unfolding ability (U) of the proteasome. (**B**–**E**) Degradation of 20 nM ubiquitinated (**B**,**C**) or nonubiquitinated (**D**,**E**) Rpn4^1–80^-PPXY-Barnase∆KL89G-C-DHFRk∆C by 100 nM WT proteasome in the presence (**B**,**D**) or absence (**C**,**E**) of 500 µM NADPH. Example gels show full-length substrate outlined in black and DHFR fragment outlined in red. Full-length (open squares) and DHFR fragment (closed squares) are shown as a percentage of total full-length present at the beginning of the reaction; full length is quantified as the sum of ubiquitinated and nonubiquitinated substrate so any deubiquitination is not misinterpreted as degradation. Dots are the results from individual experiments and error bars represent the SEM of 4–5 experiments. Dashed lines are fits in the absence of proteasome from Appendix A.

**Figure 2 biomolecules-12-00750-f002:**
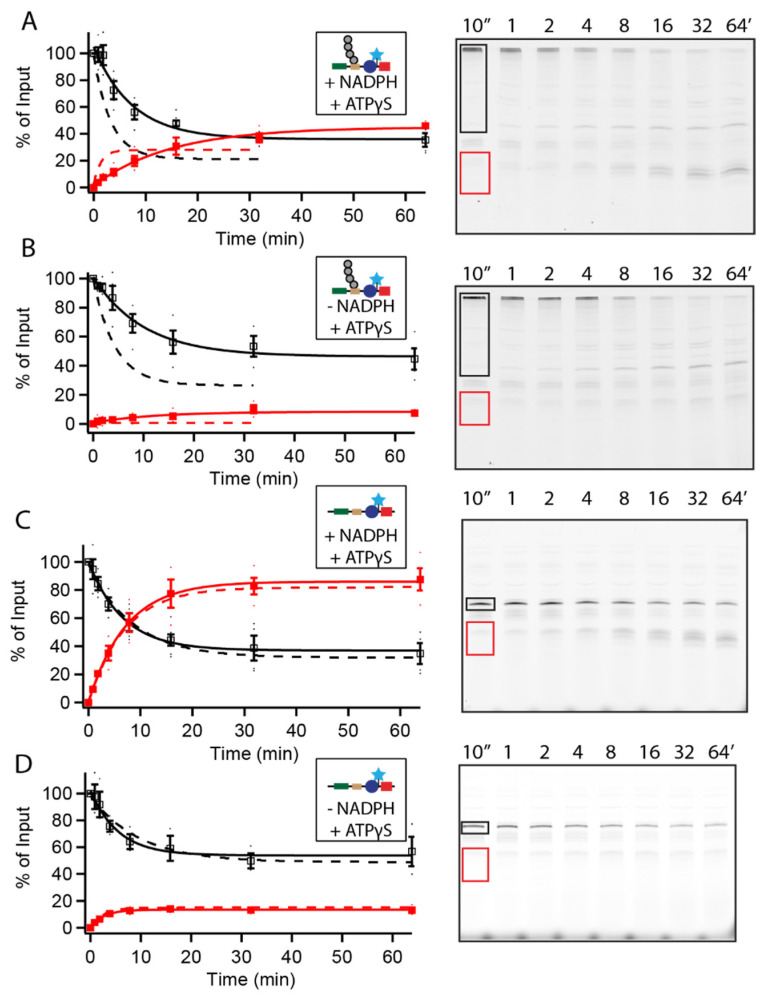
Degradation of UbID substrates is ATP-independent. (**A**–**D**) Degradation of 20 nM Rpn4^1–80^-PPXY-Barnase∆KL89G-C-DHFRk∆C by 100 nM WT proteasome in the presence of ATP-γS. Example gels show full-length substrate, (**A**,**B**) ubiquitinated or (**C**,**D**) nonubiquitinated, outlined in black, and DHFR fragment outlined in red. Full-length (open squares) and DHFR fragment (closed squares) are shown as a percentage of total full-length present at the beginning of the reaction; full length is quantified as the sum of ubiquitinated and non-ubiquitinated substrate so any deubiquitination is not misinterpreted as degradation. Dots are the results from individual experiments and error bars represent the SEM of 3–4 experiments. Dashed lines are fits in the absence of ATP-γS from Figure 1.

**Figure 3 biomolecules-12-00750-f003:**
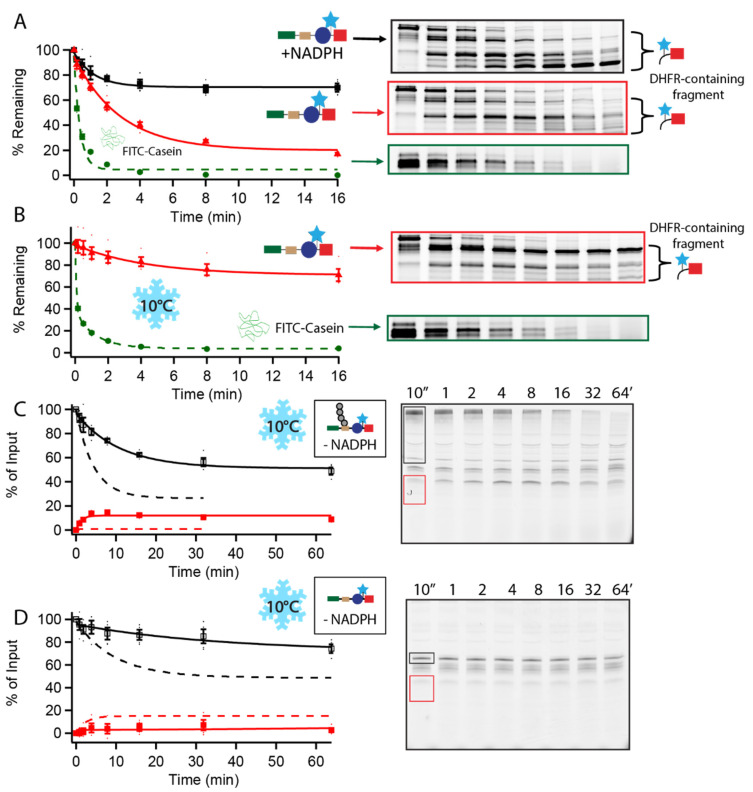
Substrate is folded in a native state and stabilized by NADPH. (**A**,**B**) Proteolysis of 80 nM Rpn4^1–80^-PPXY-Barnase∆KL89G-C-DHFRk∆C (black squares, red triangles) and 2.5 µg/mL FITC-Casein (green circles) by 0.0125 mg/mL Thermolysin at (**A**) 30 °C and (**B**) 10 °C. Example gels show full-length, nonubiquitinated substrate in the presence (outlined in black) and absence (outlined in red) of NADPH, and FITC-casein (outlined in green). Full-length substrate is shown as a percentage of total full-length present before the addition of thermolysin. Dots are the results from individual experiments and error bars represent the SEM of 4 experiments. (**C**,**D**) Degradation of 20 nM Rpn4^1–80^-PPXY-Barnase∆KL89G-C-DHFRk∆C by 100 nM WT proteasome at 10 °C. Example gels show full-length substrate, (**A**,**B**) ubiquitinated or (**C**,**D**) nonubiquitinated outlined in black, and DHFR fragment outlined in red. Full-length (open squares) and DHFR fragment (closed squares) are shown as a percentage of total full-length present at the beginning of the reaction; full length is quantified as the sum of ubiquitinated and nonubiquitinated substrate so any deubiquitination is not misinterpreted as degradation. Dots are the results from individual experiments and error bars represent the SEM of 3–4 experiments. Dashed lines are the results from identical experiments performed at 30 °C from Figure 1.

**Figure 4 biomolecules-12-00750-f004:**
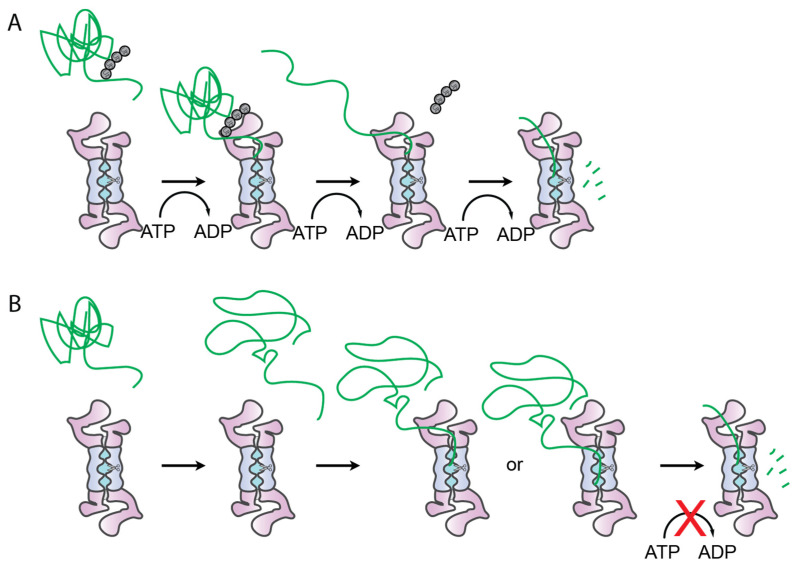
Mechanism of Ub-dependent versus UbID degradation by the proteasome. (**A**) Ubiquitin-dependent degradation occurs via ATP-dependent engagement, deubiquitination unfolding, and translocation. (**B**) Ubiquitin-independent degradation requires transient unfolding of the substrate, which is then captured by the 20S in an ATP-independent manner. It is unknown if the substrate still enters via the 19S pore.

## Data Availability

The data presented in this study are available in the article and Appendix A.

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
