# Peer review of "The 26S Proteasome Switches between ATP-Dependent and -Independent Mechanisms in Response to Substrate Ubiquitination"

_biomolecules, 2022, doi:10.3390/biom12060750_

Round 1

Reviewer 1 Report

Based on their previously reported unfolding ability assay, this group has designed unique substrates to investigate the different pathways of protein degradation in vitro. They report ubiquitin-independent degradation (UbID) is efficient with what they call “relatively unstable substrates” in an ATP-independent manner. Although ATP independent degradation of intrinsically disordered proteins was previously reported (doi:10.3390/biom10121642), the added value here is the elegant demonstration of the unfolding role in this process. 

Comments
1. the authors use the term “unstable substrates,” but they need to clarify whether they mean unstable structurally or a labile protein. 

2. NADPH, which greatly stabilizes DHFR, how? They need to provide experimental evidence or at least some references.

3. They need to describe the term “unfolding ability better.

4. An alternative interpretation is that NADPH directly inhibits the proteasome activity. The authors need to rule out this possibility in justifying their conclusions. 

5. Figure S7: mixing 20S with 19S often does not reconstitutes the 26S proteasome. You need to provide experimental evidence for your statement to support your interpretation. Why not simply incubate the reaction with the 20S alone.

Reviewer 2 Report

The paper describes the differences in proteasome-mediated degradation of recombinant proteins which were 1. in vitro ubiquitinated or 2. not-ubiquitinated and thus their degradation relies on the presence of non-ubiquitinated degron. Further, the effects of proteins folding/unfolding status and ATP availability on the degradation of the above two groups of proteins (1 and 2) are investigated. The study was performed “in tube” utilizing in vitro ubiquitination reaction and proteasome preparations, and employed proteins constructs encompassing sequences mediating both ubiquitin-dependent (UbD) and independent degradation (UbID). Although the study tackles the interesting UbID phenomenon and the experimental design is imaginative, the data analysis and presentation have some flaws, as well as at some points it is hard to agree with data interpretation proposed by Authors (Figure 2!). To improve the paper quality and to make it suitable for the publication, the following points need to be addressed:

  1. The protein MW marker is absent from the gels in paper figures as well as the MW of the analysed proteins is missing. It would be problematic to attribute MW to the fully or partially ubiquitinated constructs, however it should be possible for non-ubiqitinated proteins/degradation products and cleaved off DHFR part. Please add this information to all figures. The current form of the presentation is confusing.
  2. The putative products of engineered proteins degradation are identified solely based on the difference in protein migration speed (relative protein size). Some other identification method, like Western blot with use of anti-HIS, anti-ubiquitin or anti-DHFR antibodies, should be applied in control experiment to confirm that Cy5 bands contain the claimed proteins or their degradation products. Therefore, at least 1 gel from each experiment shown on Figures 1B, 1D, 2A, 2C should be blotted (or characteristic lanes, like time-points 0 and 32 minutes), and results added in the form of supplementary figure, parallelly with Cy5 scans to show that bands are properly assigned to the particular protein.
  3. The detailed recombinant protein composition should be provided, including the size (amino acids number and MW), and protein tags, if present in the final construct. What is the function of SUMO sequence present in the constructs?
  4. Figure 2D: The degradation of non-ubiquitinated construct is hardly visible on all 3 gels used for the quantification. Figure 2A and 2B: in the presence of ATP gamma S the band with a size of non-ubiqitinated construct appears. Then, this form accumulates, probably because it cannot be degraded by the proteasome? Therefore, based on data presented in Figure 2, concluding that ATP is dispensable for UbID is not justified. As stated in Figure 2 legend, ubiquitinated and putative non-ubiquitinated/de-ubiquitinated? form of the construct signals were quantified together. This approach is not optimal. Instead, the separate quantification, together with graph showing the kinetics for both forms of the construct should be provided, and the results properly described.
  5. Lack of the proper analysis of the differences between experimental conditions. The most important differences in UbD and UbID of the constructs should be placed on the same graph for direct comparison. The statistical analysis of data obtained at the relevant time-points should be performed.

MINOR REVISION:

  1. Figure 1, 2 and 3. Why DHFR band appears at the time point “0”? Does the construct spontaneously degrade?
  2. Line 49 – what “proteasome levels” means? Proteasome activity?
  3. Line 53 – Does Rpn4 have some target substrates? Unclear/need to be rephrased.
  4. The drawing of putative models of 19S and 20S involvement as proposed in the Discussion would be helpful for the readers.
  5. Line 280 – “sequence is not important but the structure is important” – what actually Authors mean here?
  6. Experiments with ATP gamma S. Was ATP simply replaced by ATP gamma S? Please, provide the details of the experiments.

Round 2

Reviewer 1 Report

the authors had successfully addressed my concern.

Reviewer 2 Report

Authors addressed all issues adequately. The revised manuscript can be published.